# Development and Validation of a High-Performance Liquid Chromatography with Tandem Mass Spectrometry (HPLC-MS/MS) Method for Quantification of Major Molnupiravir Metabolite (β-D-N4-hydroxycytidine) in Human Plasma

**DOI:** 10.3390/biomedicines11092356

**Published:** 2023-08-23

**Authors:** Timofey Komarov, Polina Karnakova, Olga Archakova, Dana Shchelgacheva, Natalia Bagaeva, Mariia Popova, Polina Karpova, Kira Zaslavskaya, Petr Bely, Igor Shohin

**Affiliations:** 1Center of Pharmaceutical Analytics, 8, Simferopolskiy bul, Moscow 117246, Russia; 2RUDN University, 6 Miklukho-Maklaya St., Moscow 117149, Russia; 3PROMOMED RUS, 13/1 Prospekt Mira, Moscow 129090, Russia

**Keywords:** β-D-N4-hydroxycytidine, NHC, molnupiravir, COVID-19, plasma, HPLC-MS/MS, validation, pharmacokinetics

## Abstract

Molnupiravir is an antiviral drug against viral RNA polymerase activity approved by the FDA for the treatment of COVID-19, which is metabolized to β-D-N4-hydroxycytidine (NHC) in human blood plasma. A novel method was developed and validated for quantifying NHC in human plasma within the analytical range of 10–10,000 ng/mL using high-performance liquid chromatography with tandem mass spectrometry (HPLC-MS/MS) to support pharmacokinetics studies. For sample preparation, the method of protein precipitation by acetonitrile was used, with promethazine as an internal standard. Chromatographic separation was carried out on a Shim-pack GWS C18 (150 mm × 4.6 mm, 5 μm) column in a gradient elution mode. A 0.1% formic acid solution in water with 0.08% ammonia solution (eluent A, *v*/*v*) and 0.1% formic acid solution in methanol with 0.08% ammonia solution mixed with acetonitrile in a 4:1 ratio (eluent B, *v*/*v*) were used as a mobile phase. Electrospray ionization (ESI) was used as an ionization source. The developed method was validated in accordance with the Eurasian Economic Union (EAEU) rules, based on the European Medicines Agency (EMA) and Food and Drug Administration (FDA) guidelines for the following parameters and used within the analytical part of the clinical study of molnupiravir drugs: selectivity, suitability of standard sample, matrix effect, calibration curve, accuracy, precision, recovery, lower limit of quantification (LLOQ), carryover, and stability.

## 1. Introduction

The new Coronavirus infection (Coronavirus Disease 2019 (COVID-19)) is an acute infectious disease caused by the SARS-CoV-2 virus (severe acute respiratory syndrome Coronavirus 2) that continues to pose a significant threat to public health worldwide [1]. The morbidity and mortality rates have reached extremely high values globally [2]. According to statistics, as of April 2023, over 762 million SARS-CoV-2 virus cases were confirmed worldwide, of which about 6.9 million resulted in fatalities [3,4]. Although vaccination has played a significant role in reducing viral spreading, some populations remain unvaccinated. In addition, the emergence of new variants of SARS-CoV-2 and a decrease in the effectiveness of current vaccines have led to a rise in the incidence of COVID-19 cases [5,6,7,8]. Oral antiviral therapy can prevent the progression of the disease and stop the spread of the virus [1].

Although the WHO declared the end of the pandemic on 5 May 2023, COVID-19 remains a significant health threat [9]. Therefore, the development of drugs to treat COVID-19 should not lose its urgency. COVID-19 outbreaks continue to emerge, with the disease incidence remaining unstable due to progressive mutations of SARS-CoV-2 [10].

Molnupiravir is an antiviral drug against viral RNA polymerase activity, which undergoes hydrolytic decomposition to β-D-N4-hydroxycytidine (NHC) in plasma under the influence of esterase. Subsequently, NHC is metabolized to ribonucleoside triphosphate under the action of the kinase. Ribonucleoside triphosphate is an active metabolite that, following the incorporation into the RNA of the SARS-CoV-2 virus, acts on the enzyme of RNA-dependent RNA polymerase and leads to the accumulation of mutations and suppression of viral replication [1,5,11,12,13,14,15]. The scheme of metabolic alteration of molnupiravir is shown in Figure 1. NHC does not bind to blood proteins, and is eliminated from the body by the metabolism through pyrimidine metabolic pathways [16]. Molnupiravir is rapidly absorbed from the gastrointestinal tract, and is capable of effectively crossing the hematoencephalic barrier [17]. NHC is not a substrate, an inhibitor, or an inducer of transport proteins, CYP enzymes, or P-glycoprotein [16].

Molnupiravir develops activity against Coronaviruses, including all currently known variants of SARS-CoV-2 [18]. Molnupiravir also has a high barrier to drug resistance [1]. It exhibits high oral bioavailability; therefore, molnupiravir can be administered to outpatients in the early stages of COVID-19 [12,13]. Following oral administration, molnupiravir is rapidly metabolized to the active metabolite in the blood plasma, which increases its onset of action and effectiveness [19]. The drug has been approved by the FDA (the Food and Drug Administration) for the outpatient treatment of mild to moderate COVID-19 in patients over 18 years old who are at high risk of developing serious disease [20]. Therefore, molnupiravir holds significant potential as a crucial element in combating the ongoing COVID-19 pandemic. To date, molnupiravir preparations have demonstrated good safety profiles and have a proven efficacy against new Coronavirus infections [21]. However, due to a lack of data, molnupiravir is not recommended for use in pregnant and lactating women, nor during pregnancy planning. Molnupiravir is not approved for use in children younger than 18 years of age, due to its effect on the development of bone and cartilage tissues [16].

Molnupiravir can be used both as monotherapy and in combination with other preparations. In the latter case, the antiviral activity of molnupiravir against SARS-CoV-2 was shown to increase [22]. Thus, a study into the combined application of molnupiravir with favipiravir [23] demonstrated their enhanced antiviral action, allowing molnupiravir to be used in lower doses [5]. The use of molnupiravir in combination with sotrovimab was shown to reduce the risk of severe COVID-19 outcomes [24]. The combined effects of molnupiravir with remdesivir and some other antiviral drugs were also studied [22].

The study presented in this article was conducted during the pandemic. Our aim was to develop and validate a method for quantifying molnupiravir NHC in human plasma, via high-performance liquid chromatography with tandem mass spectrometry (HPLC-MS/MS), in order to further investigate the pharmacokinetics of molnupiravir preparations. The developed and validated methodological approach was used to carry out analytical and statistical studies of molnupiravir. To the best of our knowledge, the present research was the first to address molnupiravir effects in the Russian Federation.

## 2. Materials and Methods

### 2.1. Solutions and Reagents

During the study, the following reagents were used: acetonitrile class “UHPLC Supergradient, ACS” (PanReac, Darmstadt, Germany and AppliChem, Barcelona, Spain), methanol (chemically pure, Himmed, Moscow, Russia), methanol class “HPLC gradient grade” (Himmed, Moscow, Russia), formic acid, 98% pure (PanReac, Darmstadt, Germany and AppliChem, Barcelona, Spain), 30% aqueous ammonia solution “for analysis, ACS” (PanReac, Darmstadt, Germany and AppliChem, Barcelona, Spain), and Milli-Q water (Milli-Q Integral A10, Millipore SAS, Molsheim, France). The following substances were used to prepare the stock and working standard solutions: β-D-N4-hydroxycytidine (Dezhou Hanhua Pharmaceutical Chemical Co., Ltd., Shandong, China, Lot No. 20211116, 99.12%) and promethazine hydrochloride (USP reference standard, EDQM, Strasbourg, France, Lot No. R067C0, 99.90%). Blank matrix samples of blood plasma (further referred to as blank matrix) containing no analytes or internal standards, obtained from the clinical research center, were used to prepare the samples for analysis.

### 2.2. Preparation of Stock and Working Solutions

In order to prepare the stock NHC standard solution, a standard sample was dissolved in methanol to a concentration of 200,000 ng/mL. In order to prepare the stock promethazine (internal standard, hereinafter IS PROM) standard solution, a standard sample was dissolved in acetonitrile to a concentration of 200,000 ng/mL. NHC working standard solutions were prepared by diluting the stock solution with methanol to obtain plasma concentrations corresponding to calibration levels No. 1–8 and quality control (QC) levels: LLOQ (Lower limit of quantification), L (low), M1 and M2 (middle 1, middle 2), and H (high). The IS PROM working standard solution was prepared by diluting the stock solution with acetonitrile to a concentration of 2100 ng/mL.

### 2.3. Preparation of Calibration Standards and Quality Control Samples

NHC calibration standards were prepared by adding 10 µL of working standard solutions No. 1–8 to 190 µL of blank plasma to obtain plasma concentrations of 10, 50, 100, 500, 1000, 4000, 8000, and 10,000 ng/mL, respectively. Quality control samples were prepared at LLOQ, L, M1, M2, and H levels with sample concentrations of 10, 30, 2000, 5000, and 7500 ng/mL, respectively. The concentration of IS PROM in the samples was 100 ng/mL. The samples were stored in a freezer at a temperature of −45 °C.

### 2.4. Sample Preparation

Sample preparation was carried out as follows: 10 µL of the working solution of the internal standard IS PROM was added to 200 µL of the sample (blank plasma sample, calibration standard, QC sample, and volunteer plasma sample); subsequently, 400 µL of acetonitrile was added. Following stirring on a vortex shaker for 10 s and centrifugation for 15 min with an acceleration of 15,000× *g*, the supernatant was transferred to HPLC vials and placed in an autosampler. The sample preparation scheme is depicted in Figure 2.

### 2.5. Equipment

Chromatographic separation and detection were carried out on a high-performance liquid chromatograph Nexera XR equipped with a gradient pump, a column thermostat, a degasser, an autosampler, a high-pressure flow switching valve, and a triple quadrupole mass spectrometer LCMS-8040, Shimadzu Corporation, Kyoto, Japan. Primary data processing was carried out using the LabSolutions software (Ver. 5.91), Shimadzu Corporation, Kyoto, Japan.

### 2.6. Chromatographic Conditions

Chromatographic separation was carried out on a Shim-pack GWS C18 (150 × 4.6 mm, 5 μm, Shimadzu Corporation, Kyoto, Japan) column. The Phenomenex SecurityGuard^TM^ Cartridges C18 (4 × 3.0 mm, 5 μm, Phenomenex, Torrance, CA, USA) guard column was used. The oven temperature was 40 °C. The injection volume was 10 μL. 0.1% formic acid solution in water with the addition of 0.08% ammonia solution (eluent A, *v*/*v*) and 0.1% formic acid solution in methanol with 0.08% ammonia solution mixed with acetonitrile in a 4:1 ratio (eluent B, *v*/*v*) were used as mobile phase. The mobile phase composition and its flow rate are presented in Table 1.

### 2.7. MS/MS Conditions

Electrospray ionization (ESI) was used as an ionization source. Parameters of the ionization source were as follows: nebulizer gas flow—3 L/min, drying gas flow—20 L/min, heat block temperature—400 °C, and desolvation line temperature—200 °C. The NHC and PROM analytes were detected by the positive ESI ionization mode, at the interface voltage of +5.0 kV for NHC and +4.5 kV for PROM. Detection was carried out in the mode of multiple reaction monitoring (MRM): 260.10 *m*/*z* → 128.20 *m*/*z*; 260.10 *m*/*z* → 111.10 *m*/*z* (NHC); 285.05 *m*/*z* → 198.05 *m*/*z* (PROM).

### 2.8. Validation of Analytical Method

#### 2.8.1. Selectivity

Six samples of blank plasma, two samples of blank hyperlipidemic plasma, and two samples of blank hemolysis plasma were analyzed. In addition, samples were analyzed at the LLOQ level. When analyzing blank plasma samples, the NHC and PROM signal should be less than or equal to 20% of the NHC LLOQ signal and 5% of the average PROM signals of the calibration standards and QC samples.

#### 2.8.2. Calibration Curve

The calibration curve involved 8 levels in the analytical range of 10–10,000 ng/mL. When assessing the accuracy, the obtained values of relative error (E, %) should fit into the E values ±20% of the nominal concentration for LLOQ and ±15% for the remaining levels.

#### 2.8.3. Accuracy and Precision

The analysis of plasma QC at LLOQ, L, M1, M2, and H levels was carried out within 4 sequences of 5 replicates for each NHC concentration level. The analysis was carried out during the 1st, 2nd, 3rd, and 4th sequences (intra- and inter-day). Relative standard deviation values (RSD, %) and E, % were calculated for the obtained concentrations. When assessing the accuracy, the obtained E values should be within ±20% of the nominal concentration for LLOQ and ±15% for the remaining levels, while for assessing the precision, RSD should be within ±20% of the nominal value for LLOQ and ±15% for the remaining levels.

#### 2.8.4. Lower Limit of Quantification

The LLOQ of the method is determined as the lowest amount of an NHC that can be quantitatively determined with RSD and E values of 20% or less.

#### 2.8.5. Suitability of Standard Sample

A blank plasma sample with the addition of IS PROM solution (100 ng/mL) without NHC, as well as a blank plasma sample with the addition of NHC working solution No. 8 (10,000 ng/mL) without IS, was analyzed. In addition, samples were analyzed at the LLOQ level. For a sample without NHC, the NHC signal is ≤20% of the LLOQ signal. For a sample without PROM, the PROM signal is less than 5% of the IS signal.

#### 2.8.6. Recovery

Three samples prepared from blank blood plasma, hemolysis blank plasma, and hyperlipidemic blank plasma were analyzed at each level (L, M1, M2, and H) without taking into account the recovery. In addition, identical samples were analyzed to assess the recovery. The RSD of the calculated NHC recovery from biological matrices should be less or equal to 15%.

#### 2.8.7. Matrix Effect

Six samples with the addition of NHC working standard solutions and a PROM standard solution were analyzed without taking into account the biological matrix. In addition, six samples prepared using blank plasma (blank, hyperlipidemic blank, and hemolysis blank plasma) were analyzed without taking into account the influence of the recovery of the analyzed substances and IS from the biological matrix. The RSD (CV, %) of the matrix factor normalized by the internal standard should not exceed 15%.

#### 2.8.8. Stability

Three samples were analyzed to evaluate each type of stability at the L and H levels. The obtained E values should range within ± 15%. The following types of stability were addressed:

Bench-top stability

Freshly prepared samples are analyzed at a temperature of 20 ± 5 °C.

Post-preparative stability

Samples are analyzed following storing in an autosampler at a temperature of 4 °C for 34 h.

Freeze and thaw stability

Samples that were subjected to a triple freeze–thaw cycle (36 h at a temperature of −42.5 ± 7.5 °C (freezing), 6 h at a temperature of 20 ± 5 °C (thawing)) are analyzed.

Stability of stock and working standard solutions

Samples prepared using stock and working standard solutions are analyzed following storing for 13 days at a temperature of −50 to −35 °C.

Long-term stability

Long-term stability was evaluated twice: an intermediate assessment was carried out following storing for 13 days at a temperature of −25 to −15 °C and at a temperature of −85 to −65 °C and an additional assessment following storing for 17 days at a temperature of −25 to −15 °C and at temperatures from −85 to −65 °C, since the minimum period for assessing this type of stability should correspond to the period of sample storage from the time of collection at the clinical center until the analysis of the last sample in the analytical phase of the study.

#### 2.8.9. Carryover

Sequential analysis was carried out using a calibration sample with the highest concentration (level No. 8) and human blank plasma samples. The NHC signal in blank plasma samples should be less or equal to 20% of the signal at the LLOQ level, while the IS signal of the sample should be less or equal to 5% of the IS signal.

## 3. Results

### 3.1. Method Development

Since Molnupiravir in human plasma is metabolized to β-D-N4-hydroxycytidine, which does not bind to plasma proteins [5,11,12,13], its quantitative determination as the main metabolite in human plasma is carried out [5].

Several studies have recently been published on the quantification of the NHC molnupiravir metabolite in human bodily fluids, aimed at investigating pharmacokinetic parameters. To this end, high-performance liquid chromatography with tandem mass spectrometric detection (HPLC-MS/MS), along with ultra-high-performance liquid chromatography with tandem mass spectrometric detection (UPLC-MS/MS), is used [5]. Here, sample preparation involves protein precipitation with acetonitrile, as well as ultrafiltration. The existing methods employ C18 polar columns for chromatographic separation. Both isocratic and gradient modes are used for elution. Ammonium buffer solutions are most often used as a mobile phase. Table 2 outlines published bioanalytical methods for quantifying NHC.

The conducted review showed that no published techniques for determining NHC simultaneously provide sufficient sensitivity, simplicity of sample preparation, and the most versatile eluents.

The applied conditions of mass spectrometric detection yielded NHC and PROM peaks of the highest intensity. Both positive and negative ionization modes were evaluated during the development of the mass spectrometric parameters [28]. It was observed that the signals obtained in the positive ionization mode exhibited higher intensity. Fragments obtained at different collision energies were analyzed. Subsequently, the following MRM transitions were selected: 260.10 *m*/*z* → 128.20 *m*/*z*; 260.10 *m*/*z* → 111.10 *m*/*z* (NHC); 285.05 *m*/*z* → 198.05 *m*/*z* (PROM). To improve the intensity of the NHC peak, several MRM transitions were selected. The signal-to-noise ratio of the NHC peak at the LLOQ level amounted to 15.00. The highest intensity was obtained at an ESI needle voltage of +5.0 kV for NHC and +4.5 kV for PROM.

A complete end-capped Shim-pack GWS C18 column was used for chromatographic separation. A 0.1% formic acid solution in water with the addition of 0.08% ammonia solution (*v*/*v*) and 0.1% formic acid solution in methanol with 0.08% ammonia mixed with acetonitrile solution in a 4:1 ratio were selected as the mobile phase. The optimal shape and area of chromatographic peaks were achieved when injecting 10 µL of a sample. The width of the chromatographic peak was adjusted using the gradient of the mobile phase flow rate.

The simplest method of sample preparation was used, which involved plasma protein precipitation by acetonitrile at a ratio of 1:2. This method offered optimal extraction of NHC from plasma.

Promethazine, which has a structure and physicochemical properties similar to NHC, was selected as the IS. NHC: molecular weight 259.218, logP − 2.7, pKa 12.55 [29]; promethazine: molecular weight 284.419, logP 4.29, pKa 9.05 [30].

### 3.2. Method Validation

The developed bioanalytical method was validated in accordance with the rules of bioequivalence studies within the Eurasian Economic Union [31], based on the EMA [32] and FDA [33] guidelines. The developed method underwent thorough validation for the following parameters: selectivity, suitability of standard sample, matrix effect, calibration curve, accuracy, precision, recovery, LLOQ, carryover, and stability.

#### 3.2.1. Selectivity

When analyzing blank plasma samples, the NHC and PROM signal was less or equal to 20% of the NHC LLOQ signal and 5% of the PROM signal. Selectivity meets the acceptance criteria.

#### 3.2.2. Calibration Curve

The obtained values were used to plot calibration curves in the coordinates of the ratio of analyte peak area to IS peak area from the ratio of analyte concentration to IS concentration in plasma. The calibration curves were linear. The calibration curve of one of the cycles is presented in Figure 3. The equations of calibration curves and the corresponding correlation coefficients (R) in validation cycles No. 1–5 are presented in Table 3. Weight factor: 1/C^2^. The obtained values of the correlation coefficients exceed 0.99.

#### 3.2.3. Accuracy and Precision

The obtained values of RSD for precision and E for accuracy meet the criteria of acceptance. Data on inter- and intra-day accuracy and precision for cycles No. 1–5 are presented in Table 4 and Table 5.

#### 3.2.4. Lower Limit of Quantification

The lower limit of quantification of the method amounted to 10 ng/mL. The chromatogram of human plasma containing NHC at the LLOQ level is depicted in Figure 4.

#### 3.2.5. Suitability of Standard Sample

In the sample prepared without adding the NHC solution, the NHC signal was less than 20% of the LLOQ signal. The PROM signal was less than 5% of the IS signal in a sample prepared without PROM. It was confirmed that the analyte and its impurities had no effect on the IS.

#### 3.2.6. Recovery

The average NHC recovery from various types of blank biological matrices (blank plasma, hemolyzed blank plasma, and hyperlipidemic blank plasma) equals 112.30% (RSD = 7.54%). The data are presented in Table 6.

#### 3.2.7. Matrix Effect

The matrix effect was evaluated at the L and H levels. For PROM, the matrix effect was calculated at the level of 100 ng/mL. The data are presented in Table 7. The matrix factor normalized using an internal standard was calculated on the basis of the obtained data. The RSD (CV, %) of the IS-normalized matrix factor was less than 15%.

#### 3.2.8. Stability

The resulting E values are within the normal range. The results of the stability assessment are summarized in Table 8.

#### 3.2.9. Carryover

For the samples of blank plasma, the NHC signal was within 20% of the LLOQ signal, while the IS signal amounted to ≤ 5% of the IS signal.

## 4. Conclusions

A method for the quantitative determination of the NHC molnupiravir metabolite in human plasma by HPLC-MS/MS was developed and validated. The analytical range of the method is 10–10,000 ng/mL in plasma, making it suitable for pharmacokinetic studies of molnupiravir drugs.

## Figures and Tables

**Figure 1 biomedicines-11-02356-f001:**
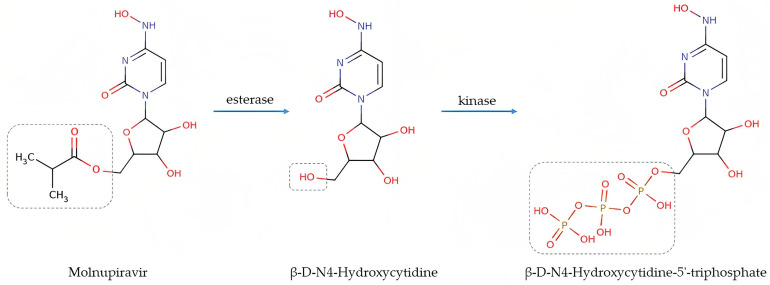
Molnupiravir metabolism.

**Figure 2 biomedicines-11-02356-f002:**
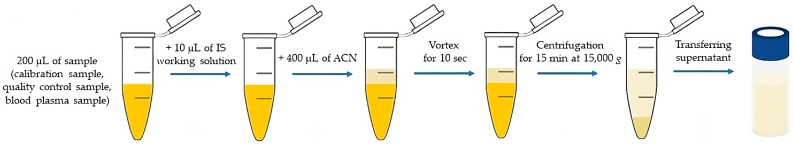
Sample preparation.

**Figure 3 biomedicines-11-02356-f003:**
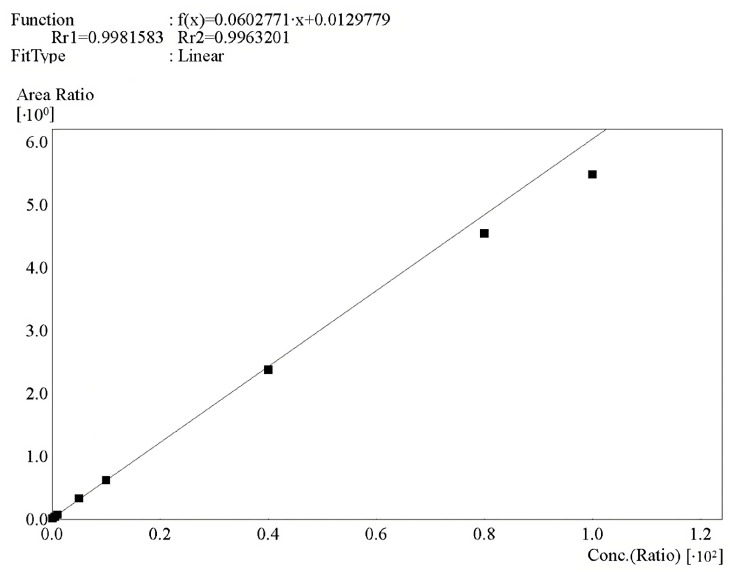
Calibration curves of NHC in the range of 10–10,000 ng/mL in human plasma.

**Figure 4 biomedicines-11-02356-f004:**
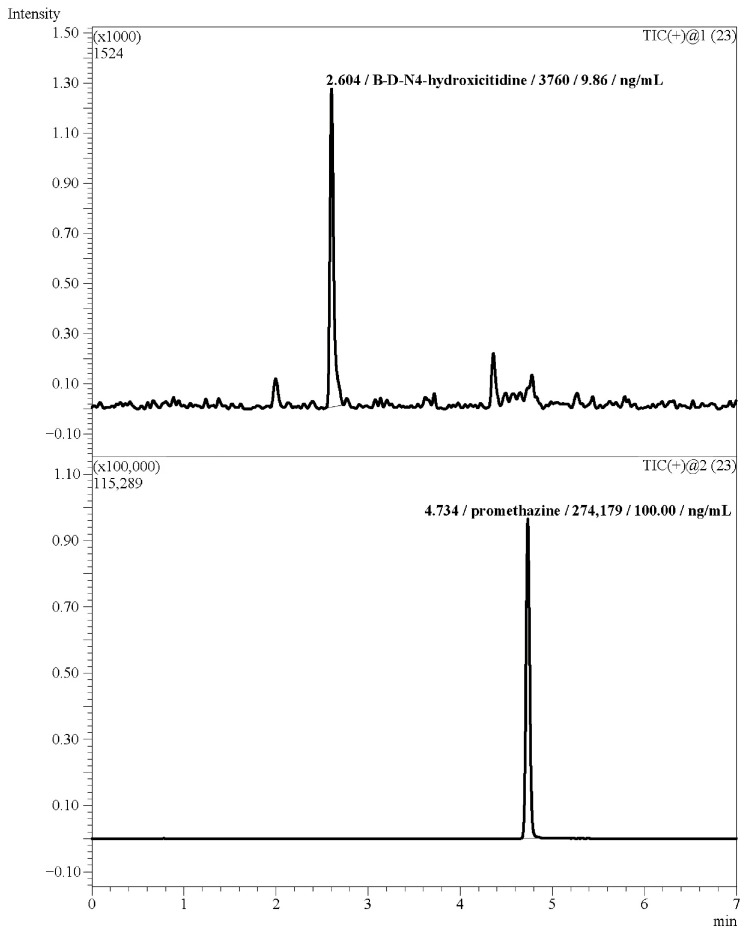
Chromatogram of a plasma sample at LLOQ level.

**Table 1 biomedicines-11-02356-t001:** Gradient elution.

Time, min	Eluent A, %	Eluent B, %	Flow Rate of Mobile Phase, mL/min
0.00	95.00	5.00	1.00
2.00	95.00	5.00	1.00
2.50	20.00	80.00	1.00
3.00	0.00	100.00	1.00
3.50	0.00	100.00	1.00
3.60	95.00	5.00	1.00
4.40	95.00	5.00	1.00
4.50	95.00	5.00	1.20
4.90	95.00	5.00	1.20
5.00	95.00	5.00	1.00
7.00	95.00	5.00	1.00

**Table 2 biomedicines-11-02356-t002:** Bioanalytical methods for NHC quantification.

Analytical Method (Ionization Source; Ionization (+/−), MRM)	Object	Sample Preparation	Column	Mobile Phase, Elution	Analytical Range, ng/mL	Ref.
HPLC-MS/MS (electrospray; +, 260.1 → 128.1)	Human plasma	Protein precipitation by ACN	Agilent Zorbax Eclipse plus C18 4.6 × 150 mm; 5 µm	0.2% CH₃COOH—MeOH, isocratic elution	20–10,000	[25]
HPLC-MS/MS (electrospray; −, (258.0 → 125.9)	Human plasma, salvia	Protein precipitation by ACN	Waters Atlantis dC18 2.1 × 100 mm; 3 µm	NH_4_CH_3_CO_2_ in H_2_O (pH = 4.3)— 1 mM NH_4_CH_3_CO_2_ in ACN, gradient elution	2.5–5000	[26]
UPLC-MS/MS (electrospray; +, 260.2 → 128.0)	Human plasma	Ultrafiltration	Scherzo SM-C18 3 × 50 mm; 3 µm	50 mM NH_4_HCO_2_:5 mM NH₄OH—80 mM NH_4_HCO_2_:8 mM NH₄OH in H_2_O:ACN [80:20], gradient elution	1–5000	[27]

**Table 3 biomedicines-11-02356-t003:** Parameters of calibration curves.

Day	Linear Equation	Correlation Coefficient (r)
1	y = 0.053x + 0.009	0.995
2	y = 0.067x + 0.007	0.997
3	y = 0.070x + 0.011	0.996
4	y = 0.111x + 0.018	0.997
5	y = 0.060x + 0.013	0.998

**Table 4 biomedicines-11-02356-t004:** Inter-day accuracy and precision of NHC determination (*n* = 5).

	Inter-Day 1 (*n* = 5)			Inter-Day 2 (*n* = 5)			Inter-Day 3 (*n* = 5)			Inter-Day 4 (*n* = 5)			Inter-Day 5 (*n* = 5)		
	Average	RSD, %	E, %	Average	RSD, %	E, %	Average	RSD, %	E, %	Average	RSD, %	E, %	Average	RSD, %	E, %
LLOQ	8.45	3.66	−15.50	10.64	9.55	6.44	9.46	15.44	−5.40	9.46	4.26	−5.42	10.01	8.56	0.12
L	28.85	9.65	−3.85	32.28	2.22	7.59	32.52	4.99	8.40	30.44	6.51	1.48	32.92	4.36	9.74
M1	2023.51	2.27	1.18	2099.69	2.68	4.98	2143.62	1.49	7.18	2184.11	0.74	9.21	1988.31	1.21	−0.58
M2	4847.18	6.88	−3.06	4974.07	1.38	−0.52	5159.72	1.68	3.19	5198.78	2.91	3.98	4738.69	4.44	−5.23
H	7007.04	2.15	−6.57	7161.09	2.76	−4.52	7046.84	1.46	−6.04	7605.53	2.63	1.41	6915.28	5.28	−7.80

**Table 5 biomedicines-11-02356-t005:** Intra-day accuracy and precision of NHC determination (*n* = 15, *n* = 20, *n* = 25).

	Intra-Day(*n* = 15)			Intra-Day (*n* = 20)			Intra-Day (*n* = 25)		
	Average	RSD, %	E, %	Average	RSD, %	E, %	Average	RSD, %	E, %
LLOQ	9.52	14.07	−4.82	9.50	12.26	−4.97	9.60	13.94	−3.95
L	31.21	7.93	4.05	31.02	7.53	3.40	31.40	7.88	4.67
M1	2088.94	3.19	4.45	2112.73	3.38	5.64	2087.85	3.19	4.39
M2	4993.66	4.61	−0.13	5044.94	4.53	0.90	4983.69	4.62	−0.33
H	7071.66	2.25	−5.71	7205.13	4.00	−3.93	7147.16	2.22	−4.70

**Table 6 biomedicines-11-02356-t006:** Assessment of NHC recovery at levels L, M1, M2, and H from various biological matrices.

Biological Matrix	Blank Plasma	Hemolyzed Blank Plasma	Hyperlipidemic Blank Plasma
L	132.76	117.59	117.17
	109.40	119.95	125.84
	120.81	106.27	124.73
M1	105.20	116.60	117.52
	106.61	120.73	104.35
	100.94	117.37	108.28
M2	105.57	118.52	98.63
	105.75	110.09	99.02
	105.57	113.89	91.38
H	114.50	112.51	113.90
	111.77	113.30	114.67
	117.51	112.90	114.90
Average	112.30		
SD	8.46		
RSD, %	7.54		

**Table 7 biomedicines-11-02356-t007:** Calculation of IS-normalized NHC matrix factors.

Biological Matrix	Blank Plasma		Hemolyzed Blank Plasma		Lipemic Blank Plasma	
	L	H	L	H	L	H
	1.89	1.29	2.68	1.39	2.45	1.50
	2.01	1.21	2.53	1.31	2.23	1.48
	1.81	1.29	2.57	1.42	2.17	1.54
	2.14	1.28	2.82	1.37	2.42	1.51
	1.77	1.30	2.69	1.37	2.23	1.60
	1.92	1.27	2.67	1.39	2.30	1.58
Average	1.92	1.27	2.66	1.38	2.30	1.53
RSD, %	6.96	2.54	3.80	2.63	4.99	2.93

**Table 8 biomedicines-11-02356-t008:** Stability assessment.

	Bench-Top		Post-Preparative		Freeze–Thaw		Long-Term 1		Long-Term 2		Stock Solution		Work Solution	
	L	H	L	H	L	H	L	H	L	H	L	H	L	H
Average	33.10	8126.07	32.33	6958.93	32.98	6938.47	32.76	6844.23	29.55	7164.52	31.02	6968.54	32.64	6947.63
E, %	10.34	8.35	7.77	−7.21	9.94	−7.49	9.21	−8.74	−1.51	−4.47	3.39	−7.09	8.79	−7.36

## Data Availability

The data presented in this study are available from the corresponding author upon special request.

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
