# Peer review of "Development and Validation of a High-Performance Liquid Chromatography with Tandem Mass Spectrometry (HPLC-MS/MS) Method for Quantification of Major Molnupiravir Metabolite (β-D-N4-hydroxycytidine) in Human Plasma"

_biomedicines, 2023, doi:10.3390/biomedicines11092356_

Round 1

Reviewer 1 Report

The Review manuscript “Development and Validation of an HPLC-MS/MS Method for 2 quantification of Major Molnupiravir Metabolite (β-D-N4-hy- 3 droxycytidine) in Human Plasma” has been written well and moderate English language. However, reviewer feels to add important information experiments to increase manuscript competency:

Author must include Deviation acceptance range for all qualifying parameters used for validation.

Author must specify the objective in the in end of introduction section to increase clarity about the manuscript work.

Author must mention nominal concentrations used in all tables.

For Validation and Pharmacokinetics, Author should refer to few suggested best papers and can use as a reference and cite in the manuscript:

Amarinder Singh et. al., (2017). Determination of ZSTK474, a novel Pan PI3K inhibitor in mouse plasma by LC–MS/MS and its application to Pharmacokinetics. Journal of Pharmaceutical and Biomedical Analysis; 149 387–393. https://www.ncbi.nlm.nih.gov/pubmed/29175554/

Vaibhav Khare, Amarinder Singh, et al. (2016). Long-circulatory nanoparticles for gemcitabine delivery: Development and investigation of pharmacokinetics and in-vivo anticancer efficacy. European Journal of Pharmaceutical Sciences; Sep 20;92:183-93 DOI. 10.1016/j.ejps.2016.07.007

Amarinder Singh et al., (2017). Re-Validation of New Develop Highly Sensitive, Simple LCMS/MS Method for the Estimation of Rohitukine and its Application in ADME/Pre-Clinical Pharmacokinetics; Mass Spectrom Purif Tech 2017, Vol 3(2): 120. DOI: 10.4172/2469- 9861.1000120.

 Table 5: Author should use same set of samples(n) while estimating Intra-day accuracy and precision of NHC. Please correct it.

Author Response

Dear Reviewer, thank you for reviewing our manuscript and for your valuable comments. Following your recommendations, we have introduced the changes described below.

  1. Deviation acceptance ranges for qualifying parameters have been added.
  2. The Introduction section has been improved by adding information about the studied drug. The aims and objectives have been specified in greater detail in a paragraph at the end of the Introduction section. Additional references have been introduced.
  3. In order to avoid overburdening the tables with numbers, we have not added nominal concentrations in the tables. Instead, the tables present the quality control levels (e.g., LLOQ, L, M1, M2, H). The NHC concentration at each quality control level is presented in section “2. 3. Preparation of calibration standards and quality control samples”.
  4. Additional references, including one from the list suggested by the Reviewer, have been added.
  5. In Table 5, the intra-day accuracy and precision values were evaluated three times: between 1–3 cycles (n=15), between 1–4 cycles (n=20), and between 1–5 cycles (n=25). In Table 4, the inter-day accuracy and precision values were assessed within each of the five validation cycles (n=5). Therefore, consistency between the sample sets in Tables 4 and 5 could not be achieved.

Reviewer 2 Report

The manuscript is well written and is appropriate for the journal.

Give the source of human plasma.

Why was a SIL internal standard not used for this work?

Italicize m/z across the manuscript.

Specify the units for x-axis on fig 3.

Tables. Please give data for 3 significant figures only.

Minor changes are required

Author Response

Dear Reviewer, thank you for reviewing our manuscript and for your valuable comments. Following your recommendations, we have introduced the changes described below.

  1. The source of human blood plasma used during method validation is presented in section “2.1 Solutions and Reagents”. Human plasma was provided by a clinical research center.
  2. We could not use a SIL internal standard due to the need to develop and validate the presented methodology, as well as to conduct the analytical part of the study, over a very short period of time. This study was conducted during the pandemic, when the SARS-COV-2 virus was actively spreading. Since no molnupiravir drug for the treatment of the new coronavirus infection was registered in Russia at that time, we faced an important task and were limited in time. The synthesis and delivery of a SIL internal standard to Russia from abroad would have taken considerable time. Therefore, the decision was made to select an internal standard from already available substances that were closest in terms of structure and/or physicochemical properties. The selected internal standard was well suited for our methodology. The method was validated; all validation parameters met the acceptance criteria.
  3. m/z has been italicized throughout the text.
  4. In Figure 3, the x-axis is the ratio of analyte concentration to internal standard concentration in plasma; therefore, the units of measurement on the x-axis are not required.
  5. Rounding to three significant figures has been applied in the tables.